# Nitrogen-plasma treated hafnium oxyhydroxide as an efficient acid-stable electrocatalyst for hydrogen evolution and oxidation reactions

Xiaofang Yang[1], Fang Zhao[2], Yao-Wen Yeh[3], Rachel S. Selinsky[1], Zhu Chen[1], Nan Yao[3], Christopher G. Tully[2], Yiguang Ju[4] & Bruce E. Koel [1]

Development of earth-abundant electrocatalysts for hydrogen evolution and oxidation reactions in strong acids represents a great challenge for developing high efficiency, durable, and cost effective electrolyzers and fuel cells. We report herein that hafnium oxyhydroxide with incorporated nitrogen by treatment using an atmospheric nitrogen plasma demonstrates high catalytic activity and stability for both hydrogen evolution and oxidation reactions in strong acidic media using earth-abundant materials. The observed properties are especially important for unitized regenerative fuel cells using polymer electrolyte membranes. Our results indicate that nitrogen-modified hafnium oxyhydroxide could be a true alternative for platinum as an active and stable electrocatalyst, and furthermore that nitrogen plasma treatment may be useful in activating other non-conductive materials to form new active electrocatalysts.

[1] Department of Chemical and Biological Engineering, Princeton University, Princeton, NJ 08544, USA. [2] Department of Physics, Princeton University, Princeton, NJ 08544, USA. [3] Princeton Institute for Science and Technology of Materials (PRISM), Princeton University, Princeton, NJ 08544, USA. [4] Department of Mechanical and Aerospace Engineering, Princeton University, Princeton, NJ 08544, USA. These authors contributed equally: Xiaofang Yang, Fang Zhao. Correspondence and requests for materials should be addressed to B.E.K. (email: bkoel@princeton.edu)

Renewable hydrogen technologies provide prospective routes to change the fossil-fuel-dominated economy through multiple high efficiency processes, specifically the production of $H_2$ from water electrolysis and generation of electricity in $H_2$ fuel cells[1]. Major impediments to the application of current water electrolyzers and fuel cells are the rates of electrochemical reactions involved, such as the hydrogen evolution reaction (HER) and hydrogen oxidation reaction (HOR), which often exhibit slow kinetics, except on platinum group metals (PGMs: Pt, Pd, and Ir), due to overpotentials for electron transfer at the electrode–electrolyte interface[2]. PGMs and their alloys are employed as active electrocatalysts due to their unique activity and durability under harsh acidic conditions but are limited in their application due to their scarcity and cost[3–5]. In contrast, under alkaline conditions, earth-abundant materials have been shown to be sufficiently stable to feasibly replace precious metals[6–8]. Although the use of high pH electrolytes addresses the need for non-precious metal electrocatalysts, other technical challenges remain. For instance, anion exchange membranes in alkaline membrane fuel cells have lower ion conductivity and suffer from more severe degradation than proton-exchange membranes (PEMs) in PEM fuel cells[9]. In electrolyzers, acidic PEM electrolysis technologies have been recognized to be superior to alkaline systems in numerous aspects, including their compact system design, lack of liquid electrolyte, high current density and energy efficiency, and high gas purity[10,11].

The search for cost-effective, earth-abundant electrocatalytic materials for PEM electrolysis with high activity and stability has identified several classes of inorganic materials that are highly active for hydrogen evolution in acidic media[12–14]. Transition metal carbides such as WC and $Mo_2C$ are promising non-precious metal alternatives to Pt for HER[15]. Their good HER activity has been attributed to the strong hybridization between metal and carbon orbitals that lead to electronic structural characteristics resembling those of Pt metal[16]. Metal chalcogenides represent a large group of inorganic HER-active materials; however, since the active sites of these materials have been shown to be located at edge sites and defects, their morphology often dictates the number of active sites, electrical conductivity, and stability[17–19]. For example, on the most well-studied metal chalcogenide for HER, i.e., two-dimensional $MoS_2$, edge S sites are active for forming sulfur hydride species that are the key intermediates for $H_2$ production[20]. Metal phosphides are another emerging class of catalysts that effectively catalyze HER[21,22]. Cobalt phosphide has demonstrated the highest HER performance in acidic media of this class[23,24]. However, despite exhibiting promising HER catalytic activity and durability in acidic media, all of these inorganic compounds are unavoidably vulnerable in practice to surface oxidation during synthesis or during exposure to air and electrolytes. Furthermore, those materials cannot be utilized to catalyze HOR in acids due to oxide formation and their intrinsic instability in strong acids at anodic potentials[25,26]. So far, electrocatalysts composed of earth-abundant elements are not comparable to PGMs for HER and HOR under acidic electrochemical conditions.

One other class of materials, transition metal oxynitrides (e.g., $MN_xO_y$, M = Ti, Ta, Hf) have been tested as catalysts for the oxygen reduction reaction[27] and oxygen evolution reaction[28]. Despite the high stability of these materials in acids, so far they have shown unsatisfactory catalytic performance for these reactions. In this paper, we report that processing Hf oxide with an atmospheric $N_2$ plasma forms an acid-insoluble hafnium oxynitride material. We propose that under electrochemical environments this material is transformed into an active oxynitride hydroxide that is capable of functioning both as a highly active HER and HOR electrocatalyst in strong acid conditions. The zero

onset potentials and high current densities demonstrate that this material is a promising alternative to Pt. This material demonstrates the excellent HOR and HER activities in acids using non-PGM catalysts, opening new opportunities to develop technologically and economically viable unitized regenerative fuel cell (URFC) systems and cost-effective PEM electrolyzers.

## Results

**Electrochemistry and electrocatalytic activity.** Hf thin films were prepared by physical vapor deposition using $Ar^+$ ion sputtering of 20 nm of Hf on both sides of roughened Au substrates that were prepared following the reported electrochemical procedure[29] and discussed further in Methods and in Supplementary Notes 1 and 2. Gold foil was chosen as a substrate because it is stable over a large potential range without showing measureable HER and HOR activity during the electrochemical polarization experiments, and roughened Au improved the physical stability of the $HfO_x$ thin films during $H_2$ generation and enabled improved characterization using surface enhanced Raman spectroscopy (SERS). The Hf film was oxidized by air exposure and then exposed to a $N_2$ plasma in a dielectric barrier discharge (DBD) reactor (Supplementary Figure 1) to incorporate nitrogen to form a hafnium oxynitride ($HfN_xO_y$) material. The surface composition and morphology (Supplementary Figure 2) of the N-modified sample were measured by X-ray photoelectron spectroscopy (XPS) and scanning electron microscopy (SEM). The samples were then electrochemically cycled over the range −200 to 1600 mV (vs reversible hydrogen electrode (RHE)) in an aqueous solution of 0.1 M $HClO_4$ prior to testing for HER and HOR performance under acidic conditions. Cyclic voltammetry (CV) using one of these $HfN_xO_y$ samples is presented in Fig. 1, along with CV data from a roughened Au foil and a similarly prepared, but without N-modification, $HfO_x$ sample for comparison. Large current at negative potentials in the cathodic scan direction for $HfN_xO_y$ is due to the HER, while the positive current in the anodic scan direction is due to the HOR. We observe a near-zero onset potential for both HER and HOR, which was formerly a unique property of PGM-based electrocatalysts. In addition, the large anodic current caused by HOR mirrors the

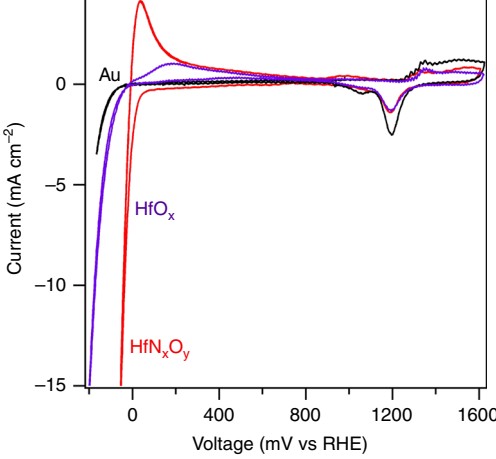

**Fig. 1** Cyclic voltammetry (CV) of N-modified Hf oxide, Hf oxide, and a roughened Au foil in acidic media hydrogen evolution reaction (HER) current from a roughened Au foil becomes visible only at −100 mV, demonstrating that the HER and hydrogen oxidation reaction activity of the $HfN_xO_y$ in the potential range of −100 to 800 mV does not arise from the exposed Au substrate. Scan rate: 50 mV s$^{-1}$ in 0.1 M $HClO_4$. The CV data are corrected for the current-resistance loss. The stable CV data shown above were from the 20th scans

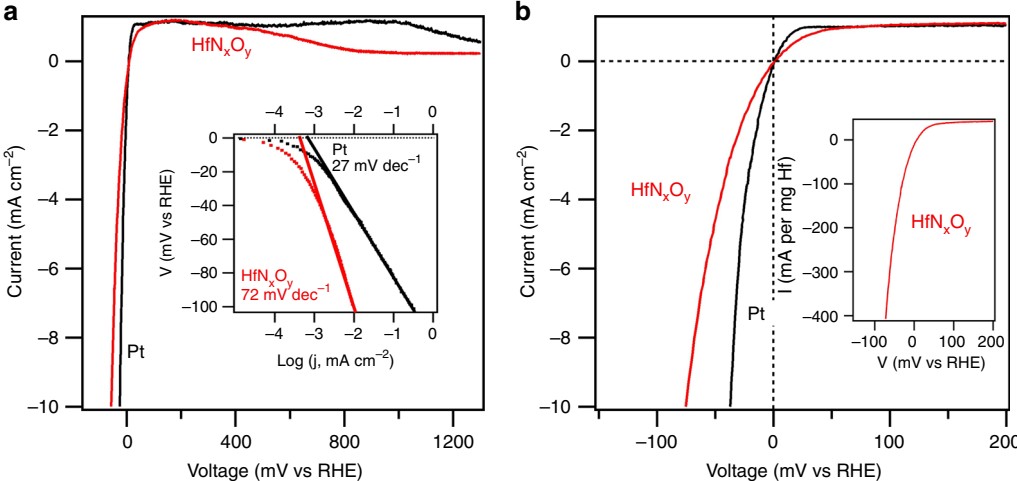

**Fig. 2** Catalytic activity of $HfN_xO_y$. **a** Comparative electrochemical behavior of Pt foil and thin film $HfN_xO_y$ electrodes in acidic media highlighting the high activity of $HfN_xO_y$ for hydrogen evolution and oxidation reactions. Inset: Tafel plots. **b** Expanded view of the activity near zero potential. Inset: Mass activity. Scan rate: $5 \, mV \, s^{-1}$ in $H_2$-purged $0.1 \, M \, HClO_4$. The non-faradaic current at $5 \, mV \, s^{-1}$ scan rate is negligible, which is shown in Supplementary Figure 5

electrochemical behavior of precious metals, such as Pt, Pd, and Ir.

At high anodic potentials, anodic peaks from 1300 to 1600 mV and cathodic peaks from 1400 to 900 mV from the $HfO_x$ and $HfN_xO_y$ thin film samples are similar to those of the roughened Au foil sample despite detection of only 0.3 at.% Au from analysis using XPS. This is indicative of incomplete conformal coating of the rough Au surface by Hf during deposition, presumably because of shadowing of some of the Au substrate from the Hf atom flux by the rough surface features. The uncoated roughened Au sample did not show any HOR activity. $HfO_x$ showed some HOR activity, but it was significantly lower than that of the $HfN_xO_y$. Supplementary Figures 3 and 4 show electrochemical analysis of other surfaces for comparison, including uncoated Au (smooth and rough) and $HfO_x$ films supported on smooth Au (both N-modified and unmodified). All these N-unmodified surfaces showed significantly lower activity than $HfN_xO_y$. Given these results, we conclude that the N-modification of $HfO_x$ by plasma treatment is critical to the observed enhanced performance of $HfN_xO_y$. Next, we will focus our discussion on the superior catalytic performance and toward determining the chemical nature of the active phase of $HfN_xO_y$.

To accurately measure the hydrogen evolution and hydrogen oxidation activity and to define the zero potential on the RHE scale, electrochemical polarization scans were carried out in $H_2$-purged $0.1 \, M \, HClO_4$ solutions and then compared to control scans using a Pt foil sample. The scanning rate was at $5 \, mV \, s^{-1}$. The HER/HOR plots shown in Fig. 2 have been corrected for the current-resistance loss from the ohmic resistance of the electrolyte obtained by impedance measurements.

Pt foil was used to benchmark HER and HOR activity as it demonstrates excellent activity in acidic media[30]. As shown in Supplementary Figure 6, a 20-nm Pt film deposited on similarly roughened Au was prepared and measured for HER and HOR activity. Although the $H_{upd}$peak area of the 20-nm Pt film sample was larger by a factor of two than that of the Pt foil sample, the Pt foil sample showed higher HER and HOR activity. Thus this Pt foil sample was used to benchmark HER and HOR activity in this study. As shown in Fig. 2a, the current polarity follows the scanning potential, switching from HER to HOR at zero potential vs RHE. Further increases in positive potential caused the HOR current to plateau due to the $H_2$ diffusion limit in the electrolyte. The current drop at higher potential (>900 mV) is caused by

surface oxidation, which removes metallic Pt sites for the reaction. Interestingly, the $HfN_xO_y$ thin film exhibits similar catalytic activity as Pt foil. The onset potential for the $HfN_xO_y$ thin film for both HER and HOR is zero, as for Pt, highlighting the excellent catalytic activity of the $HfN_xO_y$ material. In the HER potential region, the overpotential at $10 \, mA \, cm^{-2}$ is $-75 \, mV$ vs RHE, which is only 35 mV more negative than that of Pt foil. The HER activity normalized to the mass of Hf is shown as an inset in Fig. 2b and reached 400 mA per mg Hf at $-75 \, mV$. The exchange current density for HER of the $HfN_xO_y$ film reached $0.40 \, mA \, cm^{-2}$ (compared to $0.66 \, mA \, cm^{-2}$ for Pt foil) or 29 mA per mg Hf. The HER Tafel plots with steady-state current density in the range of $50 < \eta < 100 \, mV$ are shown as an inset in Fig. 2a. The Tafel slope is an important indicator of the rate-determining step of an electrochemical process. The Tafel slope of Pt was calculated to be $27 \, mV \, dec^{-1}$, consistent with previous studies[24,31]. The Tafel slope obtained for $HfN_xO_y$ was $72 \, mV \, dec^{-1}$, indicating a different reaction mechanism for HER on the $HfN_xO_y$ film. Turning our attention now to the HOR activity, we note that a polarization scan using a solid electrode in the current electrochemical set-up is not applicable for extracting the true kinetic parameters of HOR since the HOR kinetics of Pt in acid are extremely fast[32,33]. However, we still can see that the $HfN_xO_y$ film demonstrates good HOR activity, with the current reaching to the $H_2$-diffusion limited level at the overpotential of 50 mV, which is only 20 mV more positive than that of Pt foil. The most surprising observation, which differentiates this $HfN_xO_y$ film from other non-precious metal inorganic HER catalysts, is that $HfN_xO_y$ exhibits substantial HOR activity under acidic conditions. This remarkable HER and HOR activity of $HfN_xO_y$ (approaching that of Pt) under acidic conditions makes this material a true potential alternative for Pt.

**Chemical nature of the $HfN_xO_y$ thin film**. The surface composition and oxidation states of the $HfN_xO_y$ thin film was analyzed by high-resolution X-ray photoelectron spectroscopy (HRXPS). The results are summarized in Figure 3a–c and Table 1. In addition, SERS spectra of these films was obtained before and after electrochemical cycling between −200 and 1600 mV in 0.1 M $HClO_4$, as shown in Fig. 3d. The combination of HRXPS and SERS helps to elucidate the chemical nature of these $HfN_xO_y$ films and the changes induced by electrochemical measurements.

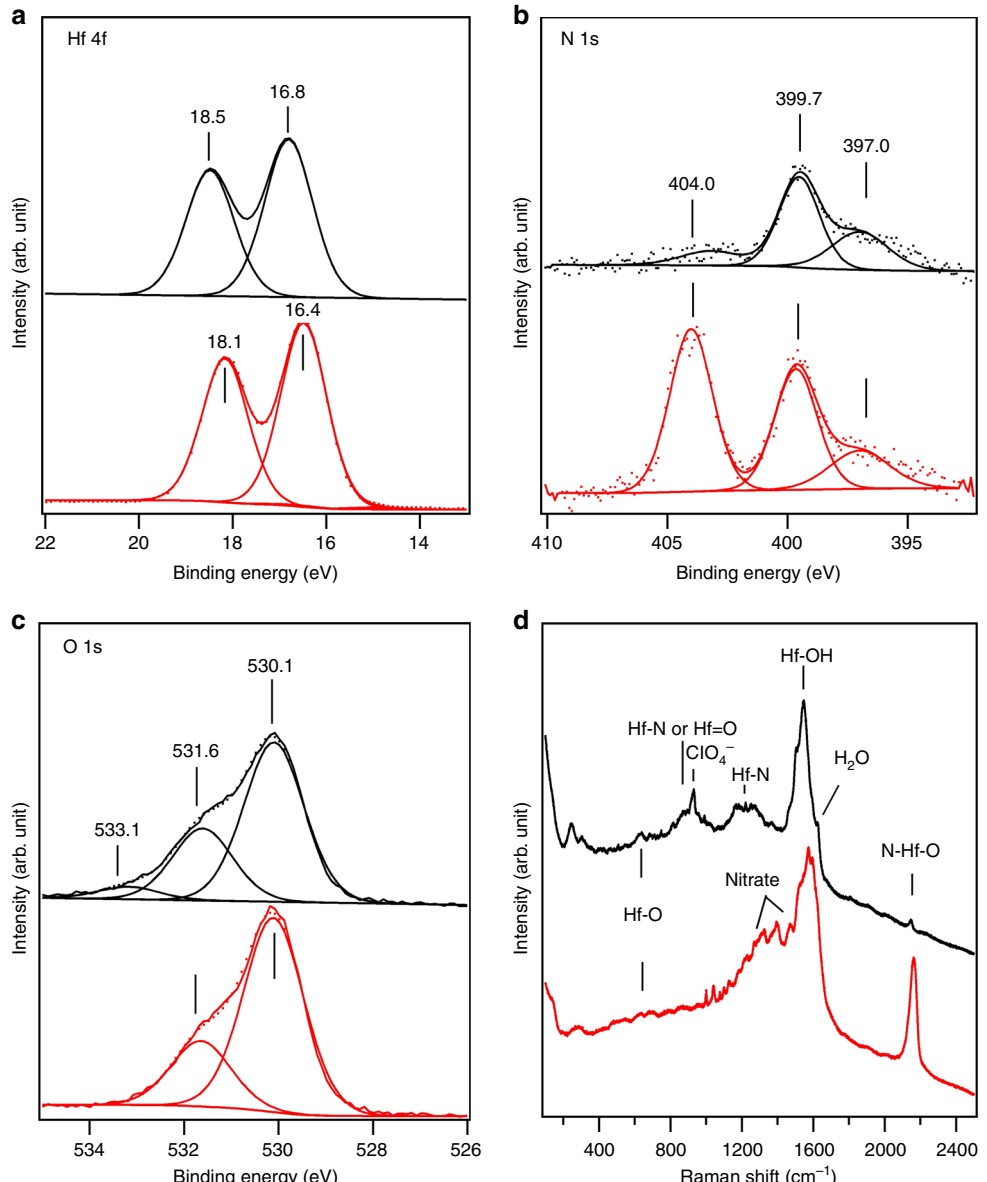

**Fig. 3** Characterization of HfN$_x$O$_y$ by X-ray photoelectron spectroscopy (XPS) and Raman spectroscopy. XPS and Raman spectra were obtained for HfN$_x$O$_y$ before (bottom or red spectra) and after (top or black spectra) electrochemical measurement (250 cyclic voltammetric (CV) scans, −200 to 1600 mV) in acidic media. XPS regional spectra are shown for: **a**, Hf 4$f$; **b**, N 1$s$; and **c**, O 1$s$. The corresponding XPS survey scan is shown in Supplementary Figure 7a. The Raman spectrum, shown in **d**, for "after CV scanning" was obtained for the wet sample after CV scans without rinsing or drying

**Table 1 Summary of the surface composition and oxidation states determined by XPS following electrochemical characterization**

| XPS peak | Au 4$f_{7/2}$ | Hf 4$f_{7/2}$ | N 1$s$ | O 1$s$ (O$^{2-}$) | O 1$s$ (OH$^-$) |
|---|---|---|---|---|---|
| Binding energy (eV) | 83.8 | 16.8 | 399.7/397.0 | 530.1 | 531.8 |
| Concentration (at. %) | 0.28 | 24.3 | 6.08 | 48.4 | 20.7 |

*XPS* X-ray photoelectron spectroscopy

In Fig. 3a–c, spectra of the HfN$_x$O$_y$ thin film are obtained before (bottom curves) and after (top curves) electrochemical measurements. Figure 3a shows a Hf 4$f_{7/2}$ peak at 16.4 eV binding energy (BE), assigned to a low oxidation state Hf$^{2+}$ initially and 16.8 eV BE (not full oxidation to Hf$^{4+}$ at 17.4 eV)[34] after electrochemical testing. Figure 3b confirmed that the N$_2$ plasma treatment led to an incorporation of N atoms into the as-grown Hf oxide film, resulting in two N 1$s$ peaks at 399.7 and 404.0 eV

BE. The high BE peak, assigned to nitrate (NO$_3^-$) species, was not stable in acid solution and was not present after electrochemical reactions. A corresponding change was observed by Raman spectroscopy, as shown in Fig. 3d, where multiple peaks in the range of 1350–1500 cm$^{-1}$ corresponding to nitrate species disappeared after electrochemical reactions. In Fig. 3b, the 399.7 eV peak is assigned to the N 1$s$ peak from oxynitride (O-Hf-N) species, which were also identified by the Raman

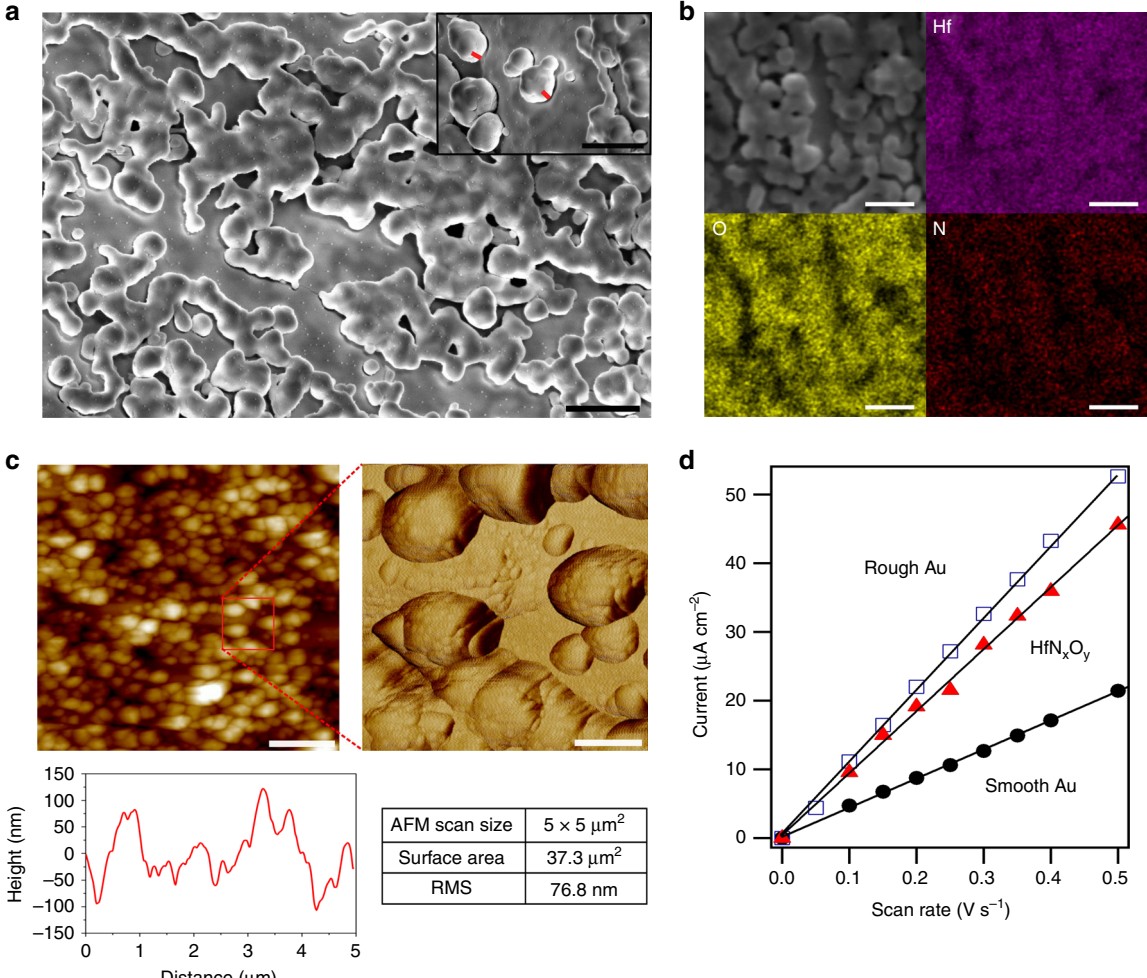

**Fig. 4** Surface characterization of a $HfN_xO_y$ film on a roughened Au surface by scanning electron microscopy (SEM) and atomic force microscopy (AFM). **a** SEM image after electrochemical measurements (250 cyclic voltammetry scans, −200 to 1600 mV) in $H_2$ purged 0.1 M $HClO_4$. Inset: 45°-tilted SEM image of the surface revealing surface morphology; height of the structures marked by red lines are 120 nm. Scale bars: 500 nm. **b** SEM image and corresponding energy-dispersive X-ray spectroscopy (EDX) images for hafnium, oxygen, and nitrogen; scale bars: 300 nm. **c** AFM mapping after electrochemical measurements showing (top left, scale bar 1 μm) a topography image and (top right, scale bar 200 nm) a phase image of the red square area and (lower left) line scan. An inset provides a table for the root mean square (RMS) roughness. **d** Double-layer charging currents of a $HfN_xO_y$ film measured at 400–500 mV (vs reversible hydrogen electrode) plotted as a function of scan rate in 0.1 M $HClO_4$. The capacitance of the $HfN_xO_y$ film was 90 μF cm$^{-2}$, which was 2.25 times larger than that obtained when using a smooth Au foil. See Supplementary Note 3 for more details on double-layer capacitance measurements

spectrum in Fig. 3d that shows an intense peak at 2160 cm$^{-1}$[35]. This Raman peak significantly decreased after electrochemical measurements, which could be due to dissolution of the hafnium oxynitride. However, the small (25%) reduction in the N 1$s$ XPS intensity at 399.7 eV indicates good stability of the N-containing material. This small reduction could be caused by the degradation of the surface oxynitride material, while the bulk of the N-containing material could be intact or transformed into other N-containing species, e.g., hydroxynitride, under electrochemical conditions. This interpretation is supported by the broad Raman peaks at 930 and 1250 cm$^{-1}$ in Fig. 3d that may be assigned to vibrations related to Hf-N species[36,37]. There is also a broad shoulder in the N 1$s$ region <400 eV that we assign to nitride-like nitrogen since peaks from metal nitrides occur near 397 eV. This nitride-like material could exist in the deeper parts of the bulk, assisting the electrical conductivity of the thin film.

The O 1$s$ peaks (Fig. 3c) can be decomposed into three components: 530.1 eV from oxidic species ($O^{2-}$), 531.6 eV from hydroxide species ($OH^-$), and 533.1 eV from adsorbed or coordinated $H_2O$. After electrochemical reaction, we observed an increase in the 531.6 eV peak intensity and assigned as corresponding to an increase in $OH^-$ concentration. The presence of Hf hydroxide was also supported by the Raman peaks at 1550–1600 cm$^{-1}$[38]. The multiple peaks in this region may be due to different local bonding environments. The atomic ratio of Hf:N determined by XPS was 4:1. It has been reported for $HfO_x$ that 25% doping with N strongly decreased (0.89 eV) the bandgap, while 50% doping with N transformed the material to a metallic oxynitride[35]. Thus we propose that one of the critical roles of N incorporation is to increase the electrical conductivity of the material, which is very important for Hf-based electrocatalysts because pristine Hf oxide is highly insulating. The broad peaks in the Raman spectra indicate that this film of $HfN_xO_y$ does not have long-range order. Thus the combination of XPS and Raman analysis indicates that the $N_2$ plasma treatment of Hf oxide incorporates N into Hf oxide (possibly oxyhydroxide) to form a material with an overall, near surface composition of roughly $Hf_1N_{0.25}O_2(OH)_{0.85}$, as determined from XPS (subject to

the well-known limitations of assuming a uniform composition throughout the probed region and thus yielding only an average composition preferentially weighted by the surface layers).

**Chemical and physical durability in strong acidic media**. Physical and chemical durability is highly important for electrocatalysts during synthesis, storage, and after regeneration using high anodic potentials. We tested the electrochemical stability of these $HfN_xO_y$ thin film electrodes in acid by CV cycling in large potential ranges (tested at both −300 to 1500 mV and −200 to 1600 mV vs RHE). These testing conditions are in stark contrast to the mild testing conditions commonly reported for non-precious metal HER catalysts such as WC, $MoS_2$, and CoP, that unlike $HfO_x$ are unable to survive high anodic potentials in strongly acidic media. The reproducibility of the CV data observed even after 1000 scans (−300 to 1500 mV, Supplementary Figures 8 and 9) indicates the high stability of these $HfN_xO_y$ electrodes in strong acid media. These results are further supported by XPS survey spectra taken before and after electrochemical testing for a new sample (250 CV cycles, −200 to 1600 mV), as shown in Supplementary Figures 2 and 7. The structural integrity of one of these $HfN_xO_y$ films was analyzed by SEM and atomic force microscopic (AFM) measurements, as shown in Fig. 4. The surface morphology was similar to that before electrochemical testing, showing features with similar sizes and shapes as the Au substrate particles. Energy-dispersive X-ray spectroscopic (EDX) mapping, as shown in Fig. 4b, enabled measurements of the distribution of Hf, O, and N in the films that showed that these elemental distributions were correlated. An SEM image obtained at a 45° tilt, as shown in the right inset of Fig. 4a, enables us to estimate the height of the Au particles to be 50–100 nm, which is consistent with the heights measured by AFM topography, as shown in Fig. 4c. The surface roughness factor (film surface area normalized to geometric area) based on the image in Fig. 4c (top left) was 1.5, which is slightly less than the electrochemical surface roughness factor of 2.25 that was determined by double-layer measurements as shown in Fig. 4d. This low surface roughness establishes that the catalytic performance of the $HfN_xO_y$ film mainly was not from an increase in the surface area caused by using the roughened Au substrate. In conclusion, the excellent chemical and physical stability of the $HfN_xO_y$ films observed here demonstrates a promising lead for developing stable electrocatalysts that can operate in strong acidic conditions.

## Discussion

In this study, we report on our measurements using a novel N-containing Hf oxide catalyst that demonstrates excellent performance for the HER and, to the best of our knowledge, an unprecedented high activity for the HOR using non-PGM catalysts under acidic conditions. We propose that a significant influence of N incorporation into Hf oxide, to promote the activity of the material, may arise from the increase in electrical conductivity of the film. Although pristine Hf oxide is well known to be highly stable in strong acids, it is an extremely insulating material and thus not useful as an electrocatalyst. $N_2$ plasma treatment is an effective approach to induce N incorporation into hafnium oxide thin films. This plasma may also create new sites or defects due to interaction with plasma species. Based on Raman spectroscopy and surface characterization by XPS, we propose that the thin film after electrochemical measurements was an N-containing oxyhydroxide.

Lack of a highly active, stable, earth-abundant electrocatalyst for carrying out HER and HOR in strongly acidic conditions is a major technical challenge for developing economical PEM-based

electrolyzers and fuel cells, and the discovery of this excellent performance by a catalyst formed by N-incorporation into Hf oxide provides a new strategy for replacing Pt metal as an electrocatalyst for HER and HOR in strong acidic media. The natural abundance of Hf is significantly higher than any of the active HER/HOR precious metals (Pt, Pd, Rh, Ru, and Ir), but the currently feasibly minable amount of Hf may not be sufficient for widespread practical use. Results reported herein have broad implications for using nitrogen incorporation (e.g., $N_2$ plasma treatment) to activate other non-conductive compounds and films to form electrocatalysts. Specifically, similar research effort should be extended to Zr-based materials since Zr belongs to the same group as Hf but is much more abundant. In addition, little is known currently about the origin of the HER/HOR activity or the active sites of these $HfN_xO_y$ thin films. Further work should explore possible support effects from the Au substrate and determine the optimal concentration of N modification for the best catalytic performance.

## Methods

**Synthesis of N-modified $HfO_x$ thin films**. The Hf thin film was deposited on a $4 \times 20$ mm$^2$ Au foil (Aldrich, 99.999%) using the VCR IBS/TM200S ion beam sputter at the PRISM Imaging and Analysis Center (IAC) at Princeton University. The sample stage was oriented perpendicular to the Hf target (ESPI, 99.9%) and rotated to obtain even Hf coverage on the Au substrates. The Ar pressure was set at $2 \times 10^{-5}$ torr during deposition and the thickness of Hf was monitored by quartz crystal microbalance. After 1 h of deposition, the Au substrate was covered by a 20-nm Hf film. Upon opening the chamber, metallic Hf quickly oxidized forming Hf oxide from exposure to air. The Hf sample was then treated by a 500-torr $N_2$ plasma for 1 h. The $N_2$ plasma was generated in a home-made DBD cell, powered by a 20-kHz, 20-kV AC power supply. The DBD plasma set-up and discharge conditions are shown in Supplementary Figure 1. After plasma treatment, the composition, oxidation states, and morphology of the samples were analyzed by a range of surface characterization techniques, including HRXPS, Raman, and SEM.

**Electrochemical measurement**. A standard three-electrode electrochemical cell was used with a graphite counter electrode and a Ag/AgCl/Sat. KCl reference electrode (Pine Research Instrument, RREF0024). We used graphite as the counter electrode to avoid possible Pt contamination that can occur when using a Pt counter electrode. During HER/HOR polarization testing, the sample was scanned from −100 to 1300 mV vs RHE at 5 mV s$^{-1}$ scan rate. This potential range was chosen to cover both HER and HOR and to assess the material stability at high anodic potentials. After electrochemical measurements in the potential range −200 to 1600 mV for 250 cycles, the samples were rinsed with pure water and the remaining water was quickly removed using a stream of nitrogen. The surface morphology, composition, and chemical states of the tested surfaces were analyzed by HRXPS, Raman, SEM, EDX, and AFM. Polarization curves are plotted with the $x$ axis scaled to the RHE. The zero potential was determined by the onset of the HER/HOR scan using a Pt electrode and $H_2$-purged 0.1 M $HClO_4$. This allows for correction of the potential drift of the Ag/AgCl reference electrode and the pH shift of the solution between different measurements. The activity was normalized to the geometrical surface area of the electrode surfaces. The electrochemical surface area was also evaluated by double-layer analysis. A fresh sample of $HfN_xO_y$ was also prepared to test for stability by scanning from −300 to 1500 mV for 1000 cycles.

**High-resolution XPS**. HRXPS analysis of the $HfO_x$ samples was conducted using a ThermoFisher K-Alpha X-ray Photoelectron Spectrometer at the PRISM IAC at Princeton University. This system is equipped with a monochromatic X-ray source and a focusing lens allowing for analysis areas from 30 to 400 μm in 5-μm steps. In this study, we selected a 400 μm X-ray spot for all XPS measurements. Survey spectra were taken at 200 eV pass energy and the high-resolution spectra for the O 1$s$, Hf 4$f$, and N 1$s$ regions were recorded at 20 eV pass energy. The XPS system has been calibrated by recording a clean Au sample with the Fermi edge BE at 0 eV and Au 4f$_{7/2}$ BE at 84.0 eV. The C 1$s$ peaks of adventitious carbon present on the samples were used to calibrate the BEs of the XPS peaks for different samples. The C 1$s$ peak position in these measurements was at 284.7 ± 0.1 eV BE. In addition, the Au signals from the substrate were also aligned together, with the Au 4f$_{7/2}$ peak at 84.0 eV BE. Film surface and chemical compositions were determined using the CasaXPS (Casa Software Ltd.) fitting software.

**High-resolution SEM (HR-SEM) and EDX**. HR-SEM, performed on a Verios 460 Extreme High Resolution SEM (XHR-SEM) at the PRISM IAC, was utilized to characterize all the thin film samples, including smooth Au, rough Au, $HfO_x$, and N-modified $HfO_x$. An Oxford EDX spectrometer, installed inside the SEM

chamber, was utilized for element mapping and analysis. The SEM images were obtained under a beam with 3–5 kV and 50 pA. We compared the surface morphologies of different electrodes by SEM (Supplementary Figure 10a–j). These surfaces included a smooth Au sample, a roughened Au sample, 20 nm HfO$_x$ film coated on smooth Au foils (modified or unmodified with N), and 20 nm HfO$_x$ film on coated on roughened Au foils (modified or unmodified with N) before and after electrochemical measurements.

**Atomic force microscopy**. A Bruker NanoMan AFM at the PRISM IAC was utilized for topographical imaging of the N-modified HfO$_x$ samples after the HER/HOR measurements in tapping mode. The surface roughness was determined by AFM mapping using a 100-μm$^2$ area.

**Raman spectroscopy**. The Raman spectra of N-modified HfO$_x$ at room temperature were measured using a Horiba LabRAM Aramis micro-Raman system, which consisted of a double monochromator combined with an optical microscope. The objective lens possessed 50 magnification. For excitation of the Raman spectra, a 532-nm line of an argon-ion laser was used at a power level of 0.6 mW. The Raman spectroscopy was used to probe the local bonding structure of the catalysts and to examine the influence of electrochemical testing on catalyst structure. Since the N-modified HfO$_x$ thin film was supported on a roughened Au substrate, SERS spectra were obtained.

## Data availability
The data that support the plots within this paper and other findings of this study are available from the corresponding author upon reasonable request.

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

## Acknowledgements
This material is based upon work supported by the National Science Foundation under Grant No. CHE-1465082. F.Z. acknowledges financial support by the Simons Foundation (#377485) and John Templeton Foundation (#58851). The electron microscopy and Raman characterization was partially supported by the National Science Foundation MRSEC program through the Princeton Center for Complex Materials (DMR-1420541)

## Author contributions
X.Y. and B.E.K. conceived the central ideas and directed the project. X.Y. and F.Z. synthesized and characterized HfN$_x$O$_y$ samples and studied their catalytic behavior in

acidic electrolyte, including the durability test. Y.-W.Y., R.S.S. and Z.C. conducted the SEM, AFM, and Raman measurements. Y.J. designed the plasma reactor. All authors contributed to the analysis of the results and commented on the manuscript.

## Additional information

**Competing interests:** The authors declare no competing interests.

**Journal Peer Review Information:** *Nature Communications* thanks the anonymous reviewers for their contribution to the peer review of this work. Peer reviewer reports are available.

