## [Peer Review File · Nature Communications]

Reviewers' comments:

Reviewer #1 (Remarks to the Author):

The authors reported nitrogen-“modified” hafnium oxide (HfN_xO_y) film on Au substrate as a bi-functional catalyst for hydrogen evolution(HER)/hydrogen oxidation reaction (HOR) in acidic media. As the authors mentioned in the introduction section of the manuscript, highly active bifunctional HER/HOR catalyst free from platinum-group-metals (PGMs) has not been reported. This manuscript can be of interest to readers in hydrogen/fuel cell community if the report, i.e., HfN_xO_y is highly active for both HOR/HER, was correct. In my opinion, some evidences are still needed to reach the conclusion at this stage. The details are written below. I cannot recommend this manuscript for publication in Nature Communications in the present form.

1. HOR activity

The authors mentioned in page 2, line 63–65:

“One other class of materials, transition metal oxynitrides (e.g. MN_xO_y, M=Ti, Ta, Hf) have been tested as catalysts for the oxygen reduction reaction (ORR)²⁴ and oxygen evolution reaction (OER)²⁵.”

However, in the reference 25, it was reported that titanium oxynitride was NOT active for OER and the current observed at high potential was from the oxidation of material itself, not OER by using mainly electrochemical techniques, X-ray photoelectron spectroscopy (XPS) and mass spectroscopy. Thus, reference 25 just denied the another previous paper that demonstrated titanium oxynitride was active for OER [reference 18 in the 25]. In my opinion, reference 25 was correct but is very boring paper reporting negative results which can be expected. I don't want to read similar negative paper after the publication of the current manuscript.

As I mentioned above, if HfN_xO_y was highly active for both HER/HOR, I can recommend this manuscript for publication. However, if the conclusion was incorrect, it will result in the waste of the resources of other researchers to publish paper similar to 25.

The catalytic activity for HER/HOR was evaluated in a half cell in this manuscript and only cyclic voltammetry was used. Besides, no details were written. The authors probably obtained results shown in Figure 2 after purging 0.1 M HClO₄ solution with H₂ gas, similar to determining the zero potential. However, the obtained currents could contain large portion of non-faradaic current from the double layer capacitance which is not relating to the catalytic reaction. For example, see Figure 4 of J. Power Sources, 182, 52-60 (2008). THE CURRENT OBTAINED UNDER N₂ OR Ar MUST BE SUBTRACTED FROM THAT UNDER H₂ TO CLARIFY THE HOR CURRENT FROM HfN_xO_y. Besides, comparing the activity of HfN_xO_y and Pt foil using current is not fair, even after the background correction as the surface area of these two samples should be different, probably the former is larger than the latter.

2. Comparison of HER/HOR activity

The authors mentioned in page 2, line 71-72;

“This material demonstrates the highest reported HOR and HER activity in acids using non-PGM catalysts,...”

However, no quantitative comparison with other results has been made. Which value was the highest? At least for HER activity, judging from overpotential at 10 mAcm⁻² or Tafel slope, the reported value from HfN_xO_y was not the highest one. Lower overpotential and Tafel slope have been reported elsewhere. Did the authors read reference 12 before citation?

3. Material

The authors proposed HfN_xO_y as nitrogen containing hafnium oxyhydroxide with an average composition of HfN_{0.25}O₂(OH)_{0.85}. But are they? From the XPS and Raman analyses, the surface

contained small amount of nitride (HfNx).

4. XPS

Figure 3a-b is very interesting if the charge calibration was performed correctly. The low Hf 4f binding energy of 16.4 eV is not strange for HfNxOy but only 0.4 eV of peak shift after 250 CV scans up to high potential of 1.3 V versus RHE in acidic media (pH is around 1) is a little bit surprising. Even after the potential cycling in acidic media, valence of hafnium was lower than 4+. As the charge calibration was performed using a clean Au sample, not on the HfNxOy, this manuscript may attract some doubtful opinions. It is better to perform charge calibration s using HfNxOy samples directly by (1) using C 1s peak or (2) Au 4f7/2 peak by depositing Au on HfNxOy.

If the valence of hafnium was truly lower than 4+, how the authors explain the results and charge neutrality of the sample?

5. XPS and Raman spectroscopy (Figure 3 b and d)

The authors assigned 930 and 1250 cm⁻¹ peaks in the Raman spectra as hafnium nitride, HfNx. Those peaks still remained after the potential cycling. However, the peak at 2160 cm⁻¹, which was assigned to O-Hf-N species in hafnium oxynitride, HfNxOy, nearly diminished after the cycling. These Raman results do not agree with the results from XPS. In N 1s region, both the broad peak at 397 eV and sharp peak at 399.7 eV, assigned to HfNx and HfNxOy, respectively remained after the cycling. The sentences from line 178 to 187 in page 5 do not explain this discrepancy at all. In general, HfNx should be dissolved or oxidized in the experimental conditions rather than HfNxOy [Powder Metall. Met. Ceram. 9, 821 (1970).], opposite to the authors' analyses. The deconvolution of N 1s peaks is necessary for the discussion. Besides, reconsideration of the assignments in both XPS and Raman spectra may be necessary.

6. Check the manuscript

The manuscript has not been checked at all and there are many typos etc. For example, "10mA cm⁻¹" in page 4, line 141

No volume and page number for reference 12 and 33

Volume and page number were incorrect, they should be 53 and 5427-5430 respectively for reference 21.

The last page is lacking in many references.

Check the manuscript before submission.

7. Resource of hafnium

The manuscript is interesting as a scientific paper if the conclusion was correct. The reported stability of HfNxOy after severe potential cycling (up to high potential in strongly acidic media) is also interesting. However, for the practical use, the resource of hafnium is too small; the world reserves (in HfO₂) is only one-order of magnitude larger than that of PGMs. The PEM electrolyzer or regenerative fuel cells with HfNxOy anodes will not be cost-effective as well considering the cost of hafnium. How will the authors address this issue in the future? Replace hafnium with zirconium? Some comments are needed.

Reviewer #2 (Remarks to the Author):

The hydrogen evolution and oxidation reactions have attracted great research attention in recent years because of their vital role in hydrogen production and fuel cells. So far Pt represents the most efficient electrocatalysts for HER and HOR, but it is high cost. It is highly desirable and imperative to develop new HER electrocatalysts with low-cost and high-performance. In this paper, the authors

reported hafnium oxyhydroxide with incorporated nitrogen demonstrates unprecedented high catalytic activity and stability for both HER and HOR in strong acidic media for earth-abundant materials. The results reported in this paper indicate that nitrogen-modified hafnium oxyhydroxide could be a true alternative for platinum as an active and stable electrocatalyst for HER and HOR. The results reported in this paper are very interesting, and I recommend this paper can be accepted after following minor revisions:

1. The surface morphology and structure of HfN_xO_y should be provided in this paper. For most of the catalysts, the surface morphology and structure are important.
2. As we all know, the catalytic durability is crucial for the catalysts. So the catalytic durability of HfN_xO_y for HER and HOR should be studied.
3. How about the effect of the composition of HfN_xO_y on catalytic activity and stability?
4. Some relevant references about HER electrocatalysis may be considered to be cited, such as *Angew. Chem. Int. Ed.* 2017, 56, 2960; *J. Am. Chem. Soc.* 2018, 140, 5118; *J. Am. Chem. Soc.* 2018, 140, 5118.

Reviewer #3 (Remarks to the Author):

I found the article titled "N₂-plasma treated hafnium oxyhydroxide as an efficient acid-stable electrocatalyst for hydrogen evolution and oxidation reactions" to be well written and researched as well as likely to influence thinking on Pt replacement research. However, I would advise that it would be appropriate for publication in *Nature Communications* only once two major issues were addressed.

I will start by listing some minor errors or things I would like to see changed/added that I found throughout the manuscript.

Manuscript:

Page 2, line 47: "Their good HER activity has been attributed to the strong hybridization between metal and carbon orbitals..." I would like to see a reference for this.

Page 2, line 65: Inconsistent use of period before or after the reference number.

Page 3, line 89: Possible missing article at beginning of sentence.

Page 3, line 105: The voltage range referring to cathodic peaks should probably be changed to 1.4 to 0.9 V.

Page 7: I found no reference to Supplementary Fig. 5.

Supplementary Information

Captions of Fig. 5, Fig. 6, and Fig. 8: 0.1V/s should read 0.1 V/s

Page 7: Fig. 7 captions refers to Fig. 5 incorrectly.

Manuscript Major Issues:

Issue 1:

There is a discrepancy in the discussion, figures, figure captions, and supplementary information on the CV scan range used for the physical durability testing. Some text and figures use the range -0.2 to 1.8 V (page 7, line 219) while others use -0.1 to 1.3 V (Sup. Fig. 8 caption), and yet others use -0.1 to 1.8 V (Sup. Fig. 6 caption), and even -0.2 to 1.3 V (Fig. 4 caption). This many discrepancies starts to put doubt in the reader's mind as to what was actually done during the stability testing.

I very much appreciate the discussion of the changes in the first 50 cycles as many researchers avoid discussing this phenomenon. However, given the discrepancies between the actual figures in Fig. 5 and Fig. 6 in the Supplementary Information, there were either at least two tests done which would alter the way the data is interpreted, or data was modified to exclude the range from 1.3 to 1.8 V during the first 50 scans. Either way, more discussion is needed to clarify how these graphs work together and the actual voltage range used needs to be reported. If more than one range was used, please explain why and in which order or make fresh samples and reproduce the data.

Issue 2:

Related to issue 1, I find that a 250 cycle CV scan is sub-par. My personal publications all use 10,000 cycles to determine cyclic stability and degradation. I believe that 1,000 cycles would be minimally sufficient to discuss physical stability in acidic media. I would like to see the scan number for the CV testing increased to 1,000.

Response to Reviewers

“N₂-plasma treated hafnium oxyhydroxide as an efficient acid-stable electrocatalyst for hydrogen evolution and oxidation reactions”

We appreciate the positive comments by the reviewers on the importance and potential impact of our manuscript. As described below, we have addressed all of the comments made by the reviewers and we detail the modifications/clarifications made to the revised manuscript. In addition, a few very minor non-scientific changes were made to correct additional small errors we found and to improve readability. We strongly feel that our revised manuscript is now suitable for publication.

Reviewer #1 (Remarks to the Author):

The authors reported nitrogen-“modified” hafnium oxide (HfN_xO_y) film on Au substrate as a bifunctional catalyst for hydrogen evolution(HER)/hydrogen oxidation reaction (HOR) in acidic media. As the authors mentioned in the introduction section of the manuscript, highly active bifunctional HER/HOR catalyst free from platinum-group-metals (PGMs) has not been reported. This manuscript can be of interest to readers in hydrogen/fuel cell community if the report, i.e., HfN_xO_y is highly active for both HOR/HER, was correct. In my opinion, some evidences are still needed to reach the conclusion at this stage. The details are written below. I cannot recommend this manuscript for publication in Nature Communications in the present form.

1. HOR activity

The authors mentioned in page 2, line 63–65:

“One other class of materials, transition metal oxynitrides (e.g. MN_xO_y, M=Ti, Ta, Hf) have been tested as catalysts for the oxygen reduction reaction (ORR)²⁴ and oxygen evolution reaction (OER)²⁵.”

However, in the reference 25, it was reported that titanium oxynitride was NOT active for OER and the current observed at high potential was from the oxidation of material itself, not OER by using mainly electrochemical techniques, X-ray photoelectron spectroscopy (XPS) and mass spectroscopy. Thus, reference 25 just denied another previous paper that demonstrated titanium oxynitride was active for OER [reference 18 in the 25]. In my opinion, reference 25 was correct but is very boring paper reporting negative results which can be expected. I don't want to read similar negative paper after the publication of the current manuscript.

As I mentioned above, if HfN_xO_y was highly active for both HER/HOR, I can recommend this manuscript for publication. However, if the conclusion was incorrect, it will result in the waste of the resources of other researchers to publish paper similar to 25.

Response:

We appreciate this concern raised by the reviewer. Since we first discovered the exceptional catalytic activity of the HfN_xO_y film for HER/HOR in April 2017, we have been challenging the results ourselves. We have by now repeated these experiments multiple times with new samples and checked for any detectable contamination of precious metals, such as Pt, Pd, Ir etc. We also carefully checked the purity of the Hf metal target used to synthesize the films. We carried out XPS measurements of the prepared films before and after electrochemical tests to probe the composition of surface layers, along with XPS sputter depth-profiling measurements of the electrochemically active sample to probe the composition of deeper

layers, and in all cases we did not detect any precious metal contamination. After all of these tests and evaluations, we feel that we have established a strong case for publishing our data and results.

The catalytic activity for HER/HOR was evaluated in a half cell in this manuscript and only cyclic voltammetry was used. Besides, no details were written. The authors probably obtained results shown in Figure 2 after purging 0.1 M HClO₄ solution with H₂ gas, similar to determining the zero potential. However, the obtained currents could contain large portion of non-faradaic current from the double layer capacitance which is not relating to the catalytic reaction. For example, see Figure 4 of J. Power Sources, 182, 52-60 (2008). THE CURRENT OBTAINED UNDER N₂ OR Ar MUST BE SUBTRACTED FROM THAT UNDER H₂ TO CLARIFY THE HOR CURRENT FROM HfN_xO_y. Besides, comparing the activity of HfN_xO_y and Pt foil using current is not fair, even after the background correction as the surface area of these two samples should be different, probably the former is larger than the latter.

Response:

The experimental details for the HER results shown in Figure 2 were stated in the main text of the submitted paper. However, for improved clarity, we have now added to the caption of Figure 2 in the revised text:

“Scan rate: 5 mV/s in H₂-purged 0.1 M HClO₄. The non-faradic current at a 5 mV/s scan rate is negligible, which is shown in Supplementary Fig. 5.”

Since the scan rate is only 5 mV/s, the non-faradaic current from the double layer capacitance is very small (as shown below).

Supplementary Fig. 5: Comparison of CVs of N-modified HfO_x thin film on roughened Au in N_2 - and H_2 -purged 0.1M HClO_4 at 5 mV/s scan rate. The non-faradaic current from the double layer capacitance is negligible for the HOR.

The HOR activity in Figure 2 and Supplementary Fig. 6b was normalized to the geometrical area. The electrochemical surface area of the HfN_xO_y sample is larger than that of the Pt foil because it was supported on a roughened Au foil. As we discussed in the submitted paper text for Figure 4d, the electrochemical surface area of the HfN_xO_y sample was about 2.25 times of that of a smooth Au foil. Therefore, we did not claim HfN_xO_y was more active than Pt for HOR, but we believe that it is important for the community that we report the observed substantial HOR activity of HfN_xO_y . Furthermore, we prepared a new sample with a 20 nm Pt thin film deposited on a similarly roughened Au foil.

In the following figures, which we have now added into the revised Supplementary Information as Supplementary Fig. 6, the CVs in N_2 -purged electrolyte and HER/HOR activity for polycrystalline Pt, a 20 nm Pt thin film deposited on roughened Au foil, and a HfN_xO_y thin film deposited on roughened Au foil are compared. The 20 nm-Pt thin film shows twice larger H_{upd} peak area than the poly-Pt. There are no H_{upd} peaks for HfN_xO_y . This CV comparison again excludes any HER-active PGM contamination in the HfN_xO_y thin film sample. Although the HER activity (normalized to the geometrical area) of the poly-Pt sample and the 20 nm Pt thin film sample were similar to each other, we found that the HOR activity of the poly-Pt sample was higher than the 20 nm Pt thin film. Therefore, in our revised manuscript, we have used the more active poly-Pt sample for comparison in the discussion of the HER/HOR activity of HfN_xO_y .

Supplementary Fig. 6: **a** CVs of N-modified HfO_x thin film (red), 20 nm Pt film on roughened Au (blue), and polycrystalline Pt (black) in N_2 -purged 0.1 M HClO_4 at 20 mV/s scan rate. **b** HER/HOR polarization scans for the samples shown in **a** in H_2 -purged 0.1 M HClO_4 at 5 mV/s scan rate.

To include this new information discussed above in our revised manuscript, the following sentences have been added to the text in our revised manuscript for the discussion of Figure 2:

“As shown in Supplementary Fig. 6, a 20 nm Pt film deposited on similarly roughened Au was prepared and measured for HER and HOR activity. Although the H_{upd} peak area of the 20 nm-Pt film sample was larger by a factor of two than that of the Pt foil sample, the Pt foil sample showed higher HER and HOR activity. This Pt foil sample was used to benchmark HER and HOR activity in this study.”

2. Comparison of HER/HOR activity

The authors mentioned in page 2, line 71-72;

“This material demonstrates the highest reported HOR and HER activity in acids using non-PGM catalysts,…”

However, no quantitative comparison with other results has been made. Which value was the highest? At least for HER activity, judging from overpotential at 10 mAcm⁻² or Tafel slope, the reported value from HfN_xO_y was not the highest one. Lower overpotential and Tafel slope have been reported elsewhere. Did the authors read reference 12 before citation?

Response:

The HER exchange current density of HfN_xO_y reached 0.40 mA cm⁻² (compared to 0.66 mA cm⁻² for poly-Pt). After it was normalized to the electrochemical area of the sample, the HER exchange current density of HfN_xO_y reached 0.17 mA cm⁻¹. If it is normalized to the number of moles of Hf, it is 2.54 mA/μmol Hf. At -75 mV vs RHE, the exchange current density is 65.7 mA/ μmol Hf. Thus, the reported activity is dependent on how the activity is normalized and this makes the comparison for the intrinsic activity difficult if the electrochemical surface areas of other studied materials in the literature are not given.

For this reason, we have now modified the sentence from our submitted manuscript “This material demonstrates the highest reported HOR and HER activity in acids using non-PGM catalysts” to the following sentence in our revised manuscript:

“This material demonstrates excellent HOR and HER activity in acids using non-PGM catalysts”.

3. Material

The authors proposed HfN_xO_y as nitrogen containing hafnium oxyhydroxide with an average composition of HfN_{0.25}O₂(OH)_{0.85}. But are they? From the XPS and Raman analyses, the surface contained small amount of nitride (HfN_x).

Response:

The average composition of the sample surface was determined by XPS. The N1s signal is weak because the XPS sensitivity factor for N1s is only 1.8 compared to that of 7.52 for Hf4f. All the N species, i.e. oxy-nitride, nitride and any other N contained compound, were included in the calculation. The XPS peak area ratio of N1s:Hf4f was 1: 16.7, so the atomic ratio of N1s/Hf4f was calculated to be 1:4.

We have now corrected this ratio in the main text of our revised paper.

4. XPS

Figure 3a-b is very interesting if the charge calibration was performed correctly. The low Hf 4f binding energy of 16.4 eV is not strange for HfN_xO_y but only 0.4 eV of peak shift after 250 CV scans up to high potential of 1.3 V versus RHE in acidic media (pH is around 1) is a little bit surprising. Even after the potential cycling in acidic media, valence of hafnium was lower than 4+. As the charge calibration was performed using a clean Au sample, not on the HfN_xO_y, this manuscript may attract some doubtful opinions. It is better to perform charge calibrations using HfN_xO_y samples directly by (1) using C 1s peak or (2) Au 4f_{7/2} peak by depositing Au on HfN_xO_y.

If the valence of hafnium was truly lower than 4+, how the authors explain the results and charge neutrality of the sample?

Response:

In our XPS measurements, we also obtained C1s peaks, which were not shown or discussed in the submitted paper. The C1s peak position in these measurements was 284.7 ± 0.1 eV binding energy. In addition, the Au signals from the substrate were also aligned together, with the Au 4f_{7/2} peak at 84.0 eV binding energy.

To be clear, we now add this information to the experimental section of our revised manuscript, which states:

“The C1s peaks of adventitious carbon present on the samples were used to calibrate the binding energies of the XPS peaks for different samples. The C1s peak position in these measurements was at 284.7 ± 0.1 eV binding energy. In addition, the Au signals from the substrate were also aligned together, with the Au 4f_{7/2} peak at 84.0 eV binding energy”.

We understand that the formula HfN_{0.25}O₂(OH)_{0.85} doesn't agree with charge neutrality if the Hf oxidation state is lower than 4+, although the binding energy of the Hf 4f peaks in XPS supports the presence of Hf^{δ+} (δ<4). We believe this inconsistency is due to the inherent limitations of XPS in determining the true composition of thin films. As a highly surface sensitive technique, XPS preferentially probes the surface layers.

To avoid this confusion to readers, we decided to remove our statement in our submitted paper: “Thus, the combination of XPS and Raman analysis indicates that the N₂-plasma treatment of Hf oxide incorporates N into Hf oxide (possibly oxyhydroxide) to form a material with a surface composition of Hf₁N_{0.25}O₂(OH)_{0.85}.”

In our revised manuscript, this has now been replaced with the following sentence:

“Thus, the combination of XPS and Raman analysis indicates that the N₂-plasma treatment of Hf oxide incorporates N into Hf oxide (possibly oxyhydroxide), to form a material with an overall, near surface composition of roughly Hf₁N_{0.25}O₂(OH)_{0.85}, as determined from XPS (subject to the well-known limitations of assuming a uniform composition throughout the probed region and thus yielding only an average composition preferentially weighted by the surface layers).”

5. XPS and Raman spectroscopy (Figure 3 b and d)

The authors assigned 930 and 1250 cm^{-1} peaks in the Raman spectra as hafnium nitride, HfN_x . Those peaks still remained after the potential cycling. However, the peak at 2160 cm^{-1} , which was assigned to O-Hf-N species in hafnium oxynitride, HfN_xO_y , nearly diminished after the cycling. These Raman results do not agree with the results from XPS. In N 1s region, both the broad peak at 397 eV and sharp peak at 399.7 eV, assigned to HfN_x and HfN_xO_y , respectively remained after the cycling. The sentences from line 178 to 187 in page 5 do not explain this discrepancy at all. In general, HfN_x should be dissolved or oxidized in the experimental conditions rather than Hf_xO_y [Powder Metall. Met. Ceram. 9, 821 (1970).], opposite to the authors' analyses. The deconvolution of N 1s peaks is necessary for the discussion. Besides, reconsideration of the assignments in both XPS and Raman spectra may be necessary.

Response:

In the paper cited by the reviewer [Powder Metall. Met. Ceram. 9, 821 (1970).], the stability of nitrides in the strong acids HCl, HNO_3 and H_2SO_4 were reported. Specially, it shows that the stability of nitrides were strongly dependent on the type of acid. Most nitrides were not stable in concentrated HCl or H_2SO_4 , while better stability was observed in concentrated HNO_3 . The stability of nitrides in the acid HClO_4 was not included in that cited paper. In our experiments, we only used 0.1 M HClO_4 , which is highly diluted from the concentrated acids tested in the cited paper, e.g. 12.4 M HCl, $d=1.19$, and 18 M H_2SO_4 , $d=1.84$. It also should be noted that those tests were conducted at near or above 100 °C, which is significantly higher than room temperature that was used in our experiments. Thus, the reaction conditions in those tests in the paper cited by the reviewer were much more severe than the conditions used in our experiments.

Our conclusions about the stability of our HfN_xO_y sample under our conditions were based on XPS analysis of N1s peaks, which showed that the peaks at 397.0 and 399.7 eV were only reduced by 25% after 250 electrochemical scans (as shown quantitatively in the table just below). However, the Raman peak for O-Hf-N was greatly reduced in intensity after electrochemical reaction. Thus, based on these results, we proposed that the small reduction in the N1s XPS peaks was due to the instability of surface Hf oxynitride species, but that the bulk HfN_xO_y material should still be intact or transformed to other species, e.g. hydroxynitride.

Areas of N1s peaks before and after CVs:

	N1s (404 eV)	N1s (399.4 eV)	N1s (397.0 eV)
Plasma treated	5669	4158	1841
Electrochemically cycled	759	2944	1725

According to the above analysis, we revised the discussion from our submitted paper to a new discussion in our revised manuscript on the chemical nature of HfN_xO_y as the following:

Before (from our submitted paper):

However, the unchanged N 1s intensity at 399.7 eV indicates good stability of the oxynitride, and thus, the drop in intensity of the 2160 cm^{-1} peak is better assigned as arising from structural changes. This

interpretation is supported by the broad peaks at 930 and 1250 cm^{-1} in Fig. 3d that can be assigned to HfN_x .^{36,37} There is also a broad shoulder in the N 1s region below 400 eV, corresponding to nitride-like nitrogen since peaks from metal nitrides occur near 397 eV. Raman shifts are more sensitive to the local bonding structure, while the N 1s XPS peak is better for measuring the oxidation state of N. We propose that the initial material was further oxidized during the electrochemical cycling and this strongly altered the Raman spectra.

After (now in our revised manuscript):

“However, the small (25%) reduction in the N 1s XPS intensity at 399.7 eV indicates good stability of the N-containing material. This small reduction could be caused by the degradation of the surface oxynitride material, while the bulk of the N-containing material could be intact or transformed into other N-containing species, e.g., hydroxynitride, under electrochemical conditions. This interpretation is supported by the broad Raman peaks at 930 and 1250 cm^{-1} in Fig. 3d that may be assigned to vibrations related to Hf-N species.^{36,37} There is also a broad shoulder in the N 1s region below 400 eV that we assign to nitride-like nitrogen since peaks from metal nitrides occur near 397 eV. This nitride-like material could exist in the deeper parts of the bulk, assisting the electrical conductivity of the thin film.”

6. Check the manuscript

The manuscript has not been checked at all and there are many typos etc. For example,

“10mA cm^{-1} ” in page 4, line 141

No volume and page number for reference 12 and 33

Volume and page number were incorrect, they should be 53 and 5427-5430 respectively for reference 21.

The last page is lacking in many references.

Check the manuscript before submission.

Response:

We regret that these errors made their way into our submitted paper. We have now rechecked our revised manuscript to correct these errors and all such others.

7. Resource of hafnium

The manuscript is interesting as a scientific paper if the conclusion was correct. The reported stability of HfN_xO_y after severe potential cycling (up to high potential in strongly acidic media) is also interesting. However, for the practical use, the resource of hafnium is too small; the world reserves (in HfO_2) is only one-order of magnitude larger than that of PGMs. The PEM electrolyzer or regenerative fuel cells with HfN_xO_y anodes will not be cost-effective as well considering the cost of hafnium. How will the authors address this issue in the future? Replace hafnium with zirconium? Some comments are needed.

Response:

The natural abundance of Hf is reported to be about 3.0-5.3 ppm in Earth's crust, which is significantly higher than any of the active HER/HOR precious metals (Pt, Pd, Rh, Ru and Ir, 0.0002-0.005 pm). However, we understand that the currently feasibly minable amount of Hf may not be sufficient for widespread practical use. We agree with the reviewer that Zr is much more plentiful than Hf, and we

believe that similar research efforts should be aimed at exploring the catalytic performance of other earth abundant materials such as Zr-based and Mo-based materials.

Accordingly, we have now added the following sentences to the last paragraph in our revised manuscript:

“The natural abundance of Hf is significantly higher than any of the active HER/HOR precious metals (Pt, Pd, Rh, Ru and Ir), but the currently feasibly minable amount of Hf may not be sufficient for widespread practical use. Results reported herein have broad implications for using nitrogen incorporation (e.g. N₂ plasma treatment) to activate other non-conductive compounds and films to form electrocatalysts. Specifically, similar research effort should be extended to Zr-based materials since Zr belongs to the same group as Hf but is much more abundant. In addition, little is known currently about the origin of the HER/HOR activity or the active sites of these HfN_xO_y thin films. Further work should explore possible support effects from the Au substrate and determine the optimal concentration of N modification for the best catalytic performance.”

Reviewer #2 (Remarks to the Author):

The hydrogen evolution and oxidation reactions have attracted great research attention in recent years because of their vital role in hydrogen production and fuel cells. So far Pt represents the most efficient electrocatalysts for HER and HOR, but it is high cost. It is highly desirable and imperative to develop new HER electrocatalysts with low-cost and high-performance. In this paper, the authors reported hafnium oxyhydroxide with incorporated nitrogen demonstrates unprecedented high catalytic activity and stability for both HER and HOR in strong acidic media for earth-abundant materials. The results reported in this paper indicate that nitrogen-modified hafnium oxyhydroxide could be a true alternative for platinum as an active and stable electrocatalyst for HER and HOR. The results reported in this paper are very interesting, and I recommend this paper can be accepted after following minor revisions:

1. The surface morphology and structure of HfN_xO_y should be provided in this paper. For most of the catalysts, the surface morphology and structure are important.

Response:

The HfN_xO_y material studied was present as a thin film only about 20 nm thick and supported on a roughened Au foil. The surface morphology of the thin film was characterized by SEM and AFM, and these results were already shown in Figure 4 of the submitted paper. We have characterized the chemical nature of this material by Raman spectroscopy and XPS, and this analysis indicates that the catalyst is Hf oxynitride. Possible active phases are Hf oxynitride or hydroxynitride, but active sites may also involve defects created by the plasma treatment. Our paper reports on our preliminary, but useful understanding of this exceptional material. To further elucidate the detailed nature of this material, reaction mechanism, and active sites, more studies including additional spectroscopic measurements such as operando FTIR, imaging measurements using high resolution TEM, and theoretical calculations would be highly useful. Such studies are planned by our group to follow up this work.

2. As we all known, the catalytic durability is crucial for the catalysts. So the catalytic durability of HfN_xO_y for HER and HOR should be studied.

Response:

We agree that the stability of electrocatalysts in acid condition is often an issue for many materials. We were aware of this issue and so did some limited testing of the durability of these HfN_xO_y materials by CV scanning of the catalysts between -0.3 to 1.5 V for 1000 cycles. These results are now shown in the revised manuscript as Supplementary Fig. 9, as shown below.

Supplementary Fig. 9: Stability testing of a fresh sample by repeated CV scans (only positive scans from the 50th, 100th, 250th, 500th, 750th and 1000th CV scan are shown). In the HER region, the current density slightly drops after 100 scans. This could be caused by the increased H_2 generated and dissolved in the solution during increasing the time of CV scanning. Dissolved H_2 influences the zero potential on the RHE scale. On the other hand, increased H_2 in solution causes a higher HOR current, and HOR currents were strongly influenced by the amount of H_2 gas dissolved in solution. Conditions: N_2 -purged 0.1 M HClO_4 , 100 mV/s. Potential range: -300 to 1500 mV. CVs are plotted using raw data without correction for the iR loss.

3. How about the effect of the composition of HfN_xO_y on catalytic activity and stability?

Response:

We agree that this is a very interesting point. However, in these experiments we did not try to vary the composition of HfN_xO_y . We only compared two cases: one without N modification and one with N modification to demonstrate the importance of the N_2 plasma treatment. In our material, the N:Hf atomic ratio was about 1:4, which was created by treatment with the N_2 plasma for 1 hour. The main goal of the N-modification was to increase the electrical conductivity of the material because HfO_x is normally an insulator, and as such is not practical as an electrocatalyst. We believe that the surface oxynitride is not likely to survive the electrochemical scans, but that the bulk oxynitride is likely still intact after

electrochemical experiments. This proposal is supported by our XPS measurements that show only a small reduction in the size of the N1s peak after CV cycling.

4. Some relevant references about HER electrocatalysis may be considered to be cited, such as *Angew. Chem. Int. Ed.* 2017, 56, 2960; *J. Am. Chem. Soc.* 2018, 140, 5118;

Response:

We thank the reviewer for this suggestion, and in our revised manuscript we have now added citations to both the “*J. Am. Chem. Soc.* 2018, 140, 5118” and “*Angew. Chem. Int. Ed.* 2017, 56, 2960” references.

Reviewer #3 (Remarks to the Author):

I found the article titled “N₂-plasma treated hafnium oxyhydroxide as an efficient acid-stable electrocatalyst for hydrogen evolution and oxidation reactions” to be well written and researched as well as likely to influence thinking on Pt replacement research. However, I would advise that it would be appropriate for publication in *Nature Communications* only once two major issues were addressed.

I will start by listing some minor errors or things I would like to see changed/added that I found throughout the manuscript.

Manuscript:

Page 2, line 47: “Their good HER activity has been attributed to the strong hybridization between metal and carbon orbitals...” I would like to see a reference for this.

Response:

This is based on the review paper: Chen, W.-F., Muckerman, J.T. & Fujita, E. Recent developments in transition metal carbides and nitrides as hydrogen evolution electrocatalysts. *Chem. Commun.* **49**, 8896-8909 (2013). In this paper, the authors state: “Experimental and theoretical investigations of these compounds both confirm that the introduction of carbon into the lattice of the early transition metals results in an expansion of the lattice constant. Density functional theory (DFT) calculations have indicated that the hybridization between metal d-orbitals and the carbon s- and p-orbitals brings about a broadening in the d-band structure, imparting characteristics resembling the d-band of Pt.” The original paper for this statement was: J. R. Kitchin, J. K. Norskov, M. A. Barteau and J. G. G. Chen, *Catal. Today*, 2005, **105**, 66.

In our revised manuscript, this paper by Kitchin, et al. is now cited as reference 16.

Page 2, line 65: Inconsistent use of period before or after the reference number.

Response: This has now been corrected in our revised manuscript.

Page 3, line 89: Possible missing article at beginning of sentence.

Response: This has now been corrected in our revised manuscript.

Page 3, line 105: The voltage range referring to cathodic peaks should probably be changed to 1.4 to 0.9 V.

Response: This has now been corrected in our revised manuscript.

Page 7: I found no reference to Supplementary Fig. 5.

Response:

Supplementary Fig. 5 present in the submitted manuscript has now been replaced in our revised Supplementary Information by Supplementary Fig.8, which is referred to in our revised manuscript in the section on p. 7 entitled “**Chemical and physical durability in strong acidic media.**”

Supplementary Information

Captions of Fig. 5, Fig. 6, and Fig. 8: 0.1V/s should read 0.1 V/s

Response: This has now been corrected in our revised manuscript.

Page 7: Fig. 7 captions refers to Fig. 5 incorrectly.

Response: This has now been corrected in our revised manuscript.

Manuscript Major Issues:

Issue 1:

There is a discrepancy in the discussion, figures, figure captions, and supplementary information on the CV scan range used for the physical durability testing. Some text and figures use the range -0.2 to 1.8 V (page 7, line 219) while others use -0.1 to 1.3 V (Sup. Fig. 8 caption), and yet others use -0.1 to 1.8 V (Sup. Fig. 6 caption), and even -0.2 to 1.3 V (Fig. 4 caption). This many discrepancies starts to put doubt in the reader's mind as to what was actually done during the stability testing.

I very much appreciate the discussion of the changes in the first 50 cycles as many researchers avoid discussing this phenomenon. However, given the discrepancies between the actual figures in Fig. 5 and Fig. 6 in the Supplementary Information, there were either at least two tests done which would alter the way the data is interpreted, or data was modified to exclude the range from 1.3 to 1.8 V during the first 50 scans. Either way, more discussion is needed to clarify how these graphs work together and the actual voltage range used needs to be reported. If more than one range was used, please explain why and in which order or make fresh samples and reproduce the data.

Response:

We thank the reviewer for bringing this important issue to our attention and we regret the confusion caused by this in our submitted paper.

In our revised manuscript, we have corrected some errors reported on the potential ranges. For fast CV scans, a potential range of -200 to 1600 mV was used. For the new stability tests on a fresh sample, a potential range of -300 to 1500 mV was used. As the reviewer noticed in our submitted paper, the reported CVs in Figure 5 and Figure 6 in the supplementary information show different scanning potential ranges. However, this was because different samples were used for these measurements. We observed that a stable CV required about 20-30 scans and this seems to be a general property of HfN_xO_y that is independent on any specific sample. The phenomenon was not strongly depended on the ending potential. For example, the figure below shows the first 30 cycles of CVs for a fresh sample.

To be clear, our revised Supplementary Information has new Figures 8 and 9 that replace Figures 5 and 6 in the older submitted paper, respectively. The CVs in Figures 8 and 9 were measured on a fresh sample.

Supplementary Fig. 8: Continuous CV scans in 0.1 M HClO_4 , 100 mV/s. The activity increased with the number of scans and approached a stable CV scan after 30 scans. Potential range: -300 to 1500 mV. The CVs are plotted using raw data without correction for the iR loss.

Issue 2:

Related to issue 1, I find that a 250 cycle CV scan is sub-par. My personal publications all use 10,000 cycles to determine cyclic stability and degradation. I believe that 1,000 cycles would be minimally sufficient to discuss physical stability in acidic media. I would like to see the scan number for the CV testing increased to 1,000.

Response:

We agree that a higher number of cycles is desirable to confirm the stability of the electrocatalyst. This is especially true for ORR and OER catalysts. For HER catalysts such as WC, MoS₂ and CoP, none of these materials can survive the high anodic potential polarization in strong acids that we used for testing HfN_xO_y. For example, we found strong degradation of WC in acid electrolyte even starting from the first CV cycle. We believe that neither MoS₂ nor CoP would survive these high anodic potentials in acids. In general, it is not necessary to test the stability of HER catalysts under such high anodic potentials. Thus, we think it is already very surprising to observe stable CVs from HfN_xO_y, and so 250 cycles of CV scans already indicate strong stability. To the best of our knowledge, there are no published CVs of WC, MoS₂ or CoP measured over a potential range similar to the one we used. For HOR catalysts, materials need to be stable in more anodic potentials. However, scanning to 1.8 V for a long time may be likely to cause unnecessary damage to the materials.

For this reason, and to address the reviewers comment, we re-prepared a fresh sample and ran 1000 CV scans over the potential range of -0.3 to 1.5 V (see the figure just below). For clarity, only the forward scans are shown.

These new results, as shown below, are given in our revised Supplementary Information manuscript as Supplementary Fig. 9 and replace the previously submitted, old Supplementary Fig. 5. We note that in the HER region, the current density slightly drops at > 100 scans. We believe that this change is due to the H₂ accumulated in the solution during the long time of these CV scans. Dissolved H₂ influences the zero potential on the RHE scale. On the other hand, more H₂ in the solution results in higher HOR currents.

Supplementary Fig. 9: Stability testing of a fresh sample by repeated CV scans (only positive scans from the 50th, 100th, 250th, 500th, 750th and 1000th CV scan are shown). In the HER region, the current density

slightly drops after 100 scans. This could be caused by the increased H_2 generated and dissolved in the solution during increasing the time of CV scanning. Dissolved H_2 influences the zero potential on the RHE scale. On the other hand, increased H_2 in the solution causes a higher HOR current, and HOR currents were strongly influenced by the amount of H_2 gas dissolved in the solution. Conditions: N_2 -purged 0.1 M $HClO_4$, 100 mV/s. Potential range: -300 to 1500 mV. The CVs are plotted using raw data without correction for the iR loss.

REVIEWERS' COMMENTS:

Reviewer #1 (Remarks to the Author):

The authors replied to the comments carefully.

The revised manuscript has been improved from the original one and therefore I can recommend it for publication in Nature Communications.

I have some comments relating to point 2 for the future studies of the authors.

As the authors mentioned, the electrochemical active surface areas of electrocatalysts are not given in most literatures.

Thus, comparison of the activity is difficult.

I think this situation is not good even for comparing the activity of several catalysts synthesized in the same laboratory.

Last weekend I have received an e-mail from ACS Nano and probably the editors had same opinion to recommend to report activity normalized to electrochemical surface area.

ACS Nano 2018, 12, 9635–9638

Some attempts to take electrochemical surface area into considerations and to report in publications may be needed for the future studies in the field of electrocatalysis.

Reviewer #2 (Remarks to the Author):

This paper has been well revised according to the comments of reviewers, and it can be accepted as it is now.

Reviewer #3 (Remarks to the Author):

The revised manuscript properly addressed each of my concerns and corrections. I have no further concerns about the paper being published in its current form.

Response to Reviewers

“Nitrogen-plasma treated hafnium oxyhydroxide as an efficient acid-stable electrocatalyst for hydrogen evolution and oxidation reactions”

We appreciate the positive comments by the reviewers on the importance and potential impact of our manuscript.

REVIEWERS' COMMENTS:

Reviewer #1 (Remarks to the Author):

The authors replied to the comments carefully.

The revised manuscript has been improved from the original one and therefore I can recommend it for publication in Nature Communications.

I have some comments relating to point 2 for the future studies of the authors.

As the authors mentioned, the electrochemical active surface areas of electrocatalysts are not given in most literatures.

Thus, comparison of the activity is difficult.

I think this situation is not good even for comparing the activity of several catalysts synthesized in the same laboratory.

Last weekend I have received an e-mail from ACS Nano and probably the editors had same opinion to recommend to report activity normalized to electrochemical surface area.

ACS Nano 2018, 12, 9635–9638

Some attempts to take electrochemical surface area into considerations and to report in publications may be needed for the future studies in the field of electrocatalysis.

Response:

We highly appreciate this comment and completely agree with this reviewer about the importance of the true electrochemical surface area. Furthermore, we also strongly recommend using similar types (high surface area powders, porous films, thin films, polycrystalline foils or single crystal surfaces) of materials when comparing the intrinsic activity of different materials. For example, the lower surface area of Pt foil should not serve as the benchmark for high surface area MoS₂ materials.

Reviewer #2 (Remarks to the Author):

This paper has been well revised according to the comments of reviewers, and it can be accepted as it is now.

No response required.

Reviewer #3 (Remarks to the Author):

The revised manuscript properly addressed each of my concerns and corrections. I have no further concerns about the paper being published in its current form.

No response required.